# TP$^2$DP$^2$: A Bayesian Mixture Model of Temporal Point Processes with Determinantal Point Process Prior

**Yiwei Dong**
Renmin University of China

**Shaoxin Ye**
Renmin University of China

**Yuwen Cao**
Renmin University of China

**Qiyu Han**
Renmin University of China

**Hongteng Xu** *
Renmin University of China

**Hanfang Yang** *
Renmin University of China

## Abstract

Asynchronous event sequence clustering aims to group similar event sequences in an unsupervised manner. Mixture models of temporal point processes have been proposed to solve this problem, but they often suffer from overfitting, leading to excessive cluster generation with a lack of diversity. To overcome these limitations, we propose a Bayesian mixture model of **T**emporal **P**oint **P**rocesses with **D**eterminantal **P**oint **P**rocess Prior (**TP$^2$DP$^2$**) and accordingly an efficient posterior inference algorithm based on conditional Gibbs sampling. Our work provides a flexible learning framework for event sequence clustering, enabling automatic identification of the potential number of clusters and accurate grouping of sequences with similar features. It is applicable to a wide range of parametric temporal point processes, including neural network-based models. Experimental results on both synthetic and real-world data suggest that our framework could produce moderately fewer yet more diverse mixture components, and achieve outstanding results across multiple evaluation metrics.

## 1 Introduction

As a powerful tool of asynchronous event sequence modeling, the temporal point process (TPP) plays a crucial role in many application scenarios [5, 7, 9, 28]. In practice, event sequences often demonstrate clustering characteristics, with certain sequences showcasing greater similarities when compared with others. For instance, event sequences of patient admissions may exhibit clustering patterns in response to specific medical treatments. Being able to accurately cluster event sequences can bring many benefits, including facilitating healthcare decision making. In recent years, researchers have built mixture models of TPPs to tackle the event sequence clustering problem [23, 22, 27]. However, these models often suffer from overfitting during training, leading to excessive cluster generation with a lack of diversity. Moreover, these methods require either manually setting the number of clusters in advance [22] or initializing a large number of clusters and gradually removing excessive clusters through hard thresholding [23, 27]. In addition, without imposing proper prior knowledge, the clusters obtained by these models may have limited diversity and cause the identifiability issue.

In this study, we propose a novel Bayesian mixture model of temporal point processes named TP$^2$DP$^2$ for event sequence clustering, imposing a determinantal point process prior to enhance the diversity of clusters and developing a universally applicable conditional Gibbs sampler-based algorithm for the model's posterior inference. As illustrated in Figure 1, TP$^2$DP$^2$ leverages the determinantal point process (DPP) as a repulsive prior for the parameters of cluster components,

---

*Corresponding authors.

Workshop on Bayesian Decision-making and Uncertainty, 38th Conference on Neural Information Processing Systems (NeurIPS 2024).

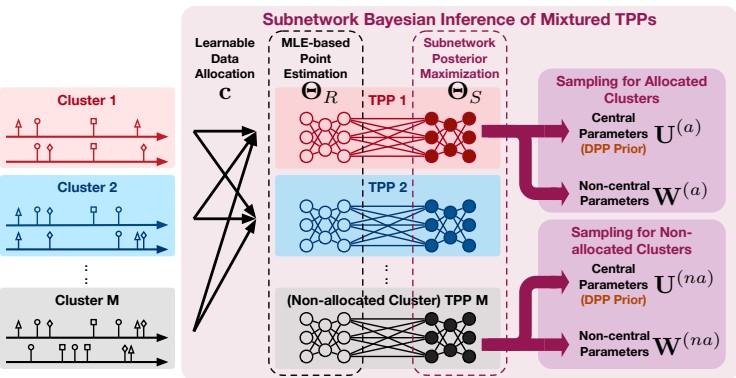

Figure 1: The pipeline of TP$^2$DP$^2$.

which contributes to generating TPPs with diverse parameters. To make TP$^2$DP$^2$ applicable for the TPPs with a large number of parameters, we apply Bayesian subnetwork inference [6], employing Bayesian inference to partially selected parameters while utilizing maximum likelihood estimation for the remaining parameters. For selected parameters, we further categorize them into central and non-central parameters, in which the central parameters mainly determine the clustering structure and thus we apply DPP priors. We design an efficient conditional Gibbs sampler-based posterior inference algorithm, in which the stochastic gradient Langevin dynamics [21] is introduced into the updating process to facilitate convergence. To our knowledge, TP$^2$DP$^2$ is the first work that explores event sequence clustering based on the TPP mixture model with DPP prior. It automatically identifies cluster numbers,with clustering results more reliable than existing variational inference methods [23, 27].

## 2 Preliminaries

**Temporal Point Processes** TPP is a kind of stochastic process that characterizes the random occurrence of events in multiple dimensions, whose realizations can be represented as event sequences, i.e., $\{(t_i, d_i)\}_{i=1}^I$, where $t_i \in [0, T]$ are time stamps and $d_i \in \mathcal{D} = \{1, ..., D\}$ are different dimensions (a.k.a. event types). Typically, we characterize a TPP by conditional intensity functions:

$$\lambda^*(t) = \sum_{d=1}^D \lambda_d^*(t), \text{ and } \lambda_d^*(t)\mathrm{d}t = \mathbb{E}[\mathrm{d}N_d(t) \mid \mathcal{H}_t]. \tag{1}$$

Here, $\lambda_d^*(t)$ is the conditional intensity function of the type-$d$ event at time $t$, $N_d(t)$ denotes the number of the occurred type-$d$ events prior to time $t$, and $\mathcal{H}_t$ denotes the historical events happening before time $t$. Given an event sequence $\boldsymbol{s} = \{(t_i, d_i)\}_{i=1}^I$, the likelihood function of a TPP can be derived based on its conditional intensity functions:

$$\mathcal{L}(\boldsymbol{s}) = \prod_{i=1}^I \lambda_{d_i}^*(t_i) \exp\Big(-\int_0^T \lambda^*(\tau)d\tau\Big). \tag{2}$$

By maximizing the likelihood in Eq. (2), we can learn the TPP model to fit the observed sequence.

**Mixture Models of TPPs** Given multiple event sequences belonging to different clusters, i.e., $\{\boldsymbol{s}_n\}_{n=1}^N$, we often leverage a mixture model of TPPs to describe their generative mechanism, leading to a hierarchical sampling process:

$$\text{1) Determine cluster: } m \sim \text{Categorical}(\boldsymbol{\pi}), \text{2) Sample sequence: } \boldsymbol{s} \sim \text{TPP}(\boldsymbol{\theta}_m), \tag{3}$$

where $\boldsymbol{\pi} = [\pi_1, ..., \pi_M] \in \Delta^{M-1}$ indicates the distribution of clusters defined on the $(M-1)$-simplex, $\text{TPP}(\boldsymbol{\theta}_m)$ is the TPP model of the $m$-th cluster, whose parameters are denoted as $\boldsymbol{\theta}_m$.

**Determinantal Point Processes** DPP [13] is a stochastic point process characterized by the unique property that its sample sets exhibit determinantal correlation. The structure of DPP is captured through a kernel function [14, 4]. Denote the kernel function by $\kappa : \mathcal{X} \times \mathcal{X} \mapsto \mathbb{R}$, where $\mathcal{X}$ represents a sample space. The density function for samples $x_1, ..., x_M \in \mathcal{X}$ in one realization of DPP is:

$$p(x_1, ..., x_M) \propto \det\{\boldsymbol{K}(x_1, ..., x_M)\}, \tag{4}$$

where $\boldsymbol{K}(x_1, ..., x_M) = [\kappa(x_i, x_j)]$ is a $M \times M$ Gram matrix corresponding to the samples. Given arbitrary two samples $x_i$ and $x_j$, we have $p(x_i, x_j) = \kappa(x_i, x_i)\kappa(x_j, x_j) - \kappa(x_i, x_j)^2 = p(x_i)p(x_j) - \kappa(x_i, x_j)^2 \leq p(x_i)p(x_j)$. Therefore, DPP manifests the repulsion between $x_i$ and $x_j$. As such, using DPP as the prior can help enhance the diversity of clustering results.

## 3  Proposed TP²DP² Model & Corresponding Posterior Inference Algorithm

The mixture model in Eq. (3) reveals that each event sequence $\boldsymbol{s}$ obeys a mixture density, i.e., $\sum_{m=1}^{M} \pi_m \mathcal{L}(\boldsymbol{s} \mid \boldsymbol{\theta}_m)$, where $M$ is a random variable denoting the number of clusters, $\boldsymbol{\pi} = [\pi_1, ..., \pi_M] \in \Delta^{M-1}$ specifies the probability of each cluster component (a TPP), and $\mathcal{L}(\boldsymbol{s} \mid \boldsymbol{\theta}_m)$ is the likelihood of the $m$-th TPP parametrized by $\boldsymbol{\theta}_m$. Given $N$ event sequences $\boldsymbol{S} = \{\boldsymbol{s}_n\}_{n=1}^{N}$, we denote cluster allocation variables of each sequence $\boldsymbol{c} = [c_1, ..., c_N] \in \{1, ..., M\}^N$, where each set $\{\boldsymbol{s}_n \mid c_n = m\}$ contains the sequences assigned to the $m$-th cluster. Accordingly, we derive the joint distribution of all variables, i.e., $p(M, \boldsymbol{\Theta}, \boldsymbol{\pi}, \boldsymbol{c}, \boldsymbol{S})$, as

$$p(M)p(\boldsymbol{\Theta} \mid M)p(\boldsymbol{\pi} \mid M) \underbrace{p(\boldsymbol{c} \mid \boldsymbol{\pi})p(\boldsymbol{S} \mid \boldsymbol{\Theta}, \boldsymbol{c})}_{\prod_{n=1}^{N} \pi_{c_n} \mathcal{L}(\boldsymbol{s}_n | \boldsymbol{\theta}_{c_n})}, \tag{5}$$

where $\boldsymbol{\Theta} = \{\boldsymbol{\theta}_m\}_{m=1}^{M} \in \mathbb{R}^P$. $p(M)$, $p(\boldsymbol{\Theta} \mid M)$ and $p(\boldsymbol{\pi} \mid M)$ are prior distributions of $M$, $\boldsymbol{\Theta}$ and $\boldsymbol{\pi}$, respectively. By Bayes theorem, the posterior $p(M, \boldsymbol{\Theta}, \boldsymbol{\pi}, \boldsymbol{c} \mid \boldsymbol{S})$ is proportional to Eq. (5).

The exact sampling from $p(M, \boldsymbol{\Theta}, \boldsymbol{\pi}, \boldsymbol{c} \mid \boldsymbol{S})$ is often intractable because the parameters of the TPPs in practice (especially those neural TPPs [15, 8, 26, 29, 16]) are too many to perform full Bayesian posterior calculation. To overcome this issue, we conduct posterior inference only on a subset of model parameters [6, 18, 12] (i.e., the "subnetwork" of the whole model). In particular, we approximate the full posterior of the TPPs' parameters $\boldsymbol{\Theta}$ as

$$p(\boldsymbol{\Theta} \mid \boldsymbol{S}) \approx p(\boldsymbol{\Theta}_S \mid \boldsymbol{S})\,\delta(\boldsymbol{\Theta}_R - \widehat{\boldsymbol{\Theta}}_R) = p(\boldsymbol{U} \mid \boldsymbol{S})p(\boldsymbol{W} \mid \boldsymbol{S})\delta(\boldsymbol{\Theta}_R - \widehat{\boldsymbol{\Theta}}_R), \tag{6}$$

where we split the model parameters $\boldsymbol{\Theta}$ into two parts, i.e., $\boldsymbol{\Theta}_S$ and $\boldsymbol{\Theta}_R$, respectively. $\boldsymbol{\Theta}_S = \{\boldsymbol{\theta}_{S,m}\}_{m=1}^{M}$ corresponds to the subnetworks of the TPPs in the mixture model, while $\boldsymbol{\Theta}_R = \{\boldsymbol{\theta}_{R,m}\}_{m=1}^{M}$ denotes the remaining parameters. In Eq. (6), $p(\boldsymbol{\Theta} \mid \boldsymbol{S})$ is decomposed into the posterior of the subnetworks $p(\boldsymbol{\Theta}_S \mid \boldsymbol{S})$ and a Dirac delta function on the remaining parameters $\boldsymbol{\Theta}_R$, in which $\boldsymbol{\Theta}_R$ is estimated by their point estimation $\widehat{\boldsymbol{\Theta}}_R = \{\widehat{\boldsymbol{\theta}}_{R,m}\}_{m=1}^{M}$, e.g., the maximum likelihood estimation achieved by stochastic gradient descent.

Unlike existing work, in Eq. (6), we further decompose the parameters in the subnetworks into two parts, i.e., $\boldsymbol{\Theta}_S = \{\boldsymbol{U}, \boldsymbol{W}\}$, where $\boldsymbol{U} = \{\boldsymbol{\mu}_m\}_{m=1}^{M}$ and $\boldsymbol{W} = \{\boldsymbol{w}_m\}_{m=1}^{M}$, respectively. For the $m$-th TPP in the mixture model, $\boldsymbol{\mu}_m$ corresponds to the "central" parameters of their conditional intensity functions, which significantly impacts the overall dynamics of event occurrence (e.g. the base intensity of Hawkes process [11]). Accordingly, the other "non-central" parameters in each subnetwork are denoted as $\boldsymbol{w}_m$, which are contingent upon specific architectures of different models. Imposing the conditional independence on the central and non-central parameters, i.e., $p(\boldsymbol{\Theta}_S | M) = p(\boldsymbol{U} | M)p(\boldsymbol{W} | M)$, we have

$$p(M, \boldsymbol{\Theta}, \boldsymbol{\pi}, \boldsymbol{c} | \boldsymbol{S}) \propto p(M)p(\boldsymbol{U} | M)p(\boldsymbol{W} | M)p(\boldsymbol{\pi} | M) \prod_{n=1}^{N} \pi_{c_n} \mathcal{L}(\boldsymbol{s}_n | \boldsymbol{\theta}_{S,c_n}, \widehat{\boldsymbol{\theta}}_{R,c_n}), \tag{7}$$

where $\widehat{\boldsymbol{\theta}}_{R,c_n}$ denotes the point estimates of the remaining parameters in the $c_n$-th TPP. The DPP prior $p(\boldsymbol{U} \mid M)$ is introduced to the central parameter $\boldsymbol{U}$ to mitigate the overfitting problem and diversify the cluster result. The computational method of DPP prior construction is introduced in Appendix. $p(\boldsymbol{W} \mid M) = \prod_{m=1}^{M} p(\boldsymbol{w}_m)$ is the prior of non-central parameters which can be Gaussian. For prior of $\boldsymbol{\pi}$, instead of directly sampling $\{\pi_m\}_{m=1}^{M}$ from its posterior distribution, we apply the ancillary variable method [3, 1] to make the posterior calculation tractable for the mixture weights $\{\pi_m\}_{m=1}^{M}$. Consider $\boldsymbol{r} = [r_1, ..., r_M]$, which consists of i.i.d. positive continuous random variables following the Gamma distribution $\Gamma(1, 1)$, each $r_m$ is independent of $M$ and $\boldsymbol{r}$ is independent of $\{\boldsymbol{U}, \boldsymbol{W}\}$. Defining $t = \sum_{m=1}^{M} r_m$ and $\boldsymbol{\pi} = [r_1/t, ..., r_M/t]$, we establish a one-to-one correspondence between $\boldsymbol{\pi}$ and $(\boldsymbol{r}, t)$. By introducing an extra random variable $v \sim \Gamma(N, 1)$, we define the ancillary variable $u = v/t$, with $p(u) = \frac{u^{N-1}}{\Gamma(N)} \int_0^{\infty} t^N e^{-ut} p(t) \mathrm{d}t$. Introducing $u$ makes

---

**Algorithm 1** Conditional Gibbs Sampler for TP$^2$DP$^2$

---

**Input**: Event sequences $\boldsymbol{S}$, priors, initialization of the cluster number, maximum number of iteration $\mathbf{T}$, number of burn-in, step sizes for each update, point estimates $\widehat{\boldsymbol{\Theta}}_R$.
**Output**: Posterior samples of variables in the model $\{M, \boldsymbol{U}, \boldsymbol{r}, \boldsymbol{W}, \boldsymbol{c}\}$.

1: Initialize parameters and set $j = 0$.
2: **while** convergence not reached and $j < \mathbf{T}$ **do**
3:      Sample non-allocated variables $(\boldsymbol{U}^{(na)}, \boldsymbol{r}^{(na)}, \boldsymbol{W}^{(na)})$ using collapsed Gibbs sampler. The sampling for $\boldsymbol{U}^{(na)}$ is given by:

$$p(\boldsymbol{U}^{(na)} \mid \boldsymbol{U}^{(a)}, \boldsymbol{r}^{(a)}, \boldsymbol{W}^{(a)}, \boldsymbol{c}, u, \boldsymbol{S}) \propto p(\boldsymbol{U}^{(a)} \cup \boldsymbol{U}^{(na)})\psi(u)^l, \text{ where } \psi(u) \text{ denotes the}$$

     Laplace transform of $p(r_m)$, i.e. $\psi(u)^l = \int \prod_{m=1}^{l} \exp(-u r_m^{(na)}) p(r_m^{(na)}) \mathrm{d}\boldsymbol{r}^{(na)}$

     The $\boldsymbol{r}^{(na)}$ and $\boldsymbol{W}^{(na)}$ is given by:

$$p(\boldsymbol{r}^{(na)} \mid \cdots) \propto \prod_{m=1}^{l} p(r_m^{(na)}) e^{-u r_m^{(na)}}, \; p(\boldsymbol{W}^{(na)} \mid \cdots) \propto \prod_{m=1}^{l} p(\boldsymbol{w}_m^{(na)}).$$

4:      Sample allocated variables $(\boldsymbol{U}^{(a)}, \boldsymbol{r}^{(a)}, \boldsymbol{W}^{(a)})$.

$$p(\boldsymbol{U}^{(a)} \mid \cdots) \propto p(\boldsymbol{U}^{(a)} \cup \boldsymbol{U}^{(na)}) \prod_{m=1}^{k} \prod_{i:c_i=m} \mathcal{L}(\boldsymbol{s}_i \mid (\boldsymbol{\mu}_m^{(a)}, \boldsymbol{w}_m^{(a)}, \widehat{\boldsymbol{\theta}}_{R,m}),$$

$$p(\boldsymbol{r}^{(a)} \mid \cdots) \propto \prod_{m=1}^{k} p(r_m^{(a)}) (r_m^{(a)})^{n_m} \exp(-u r_m^{(a)}).$$

$$p(\boldsymbol{W}^{(a)} \mid \cdots) \propto \prod_{m=1}^{k} p(\boldsymbol{w}_m^{(a)}) \prod_{i:c_i=m} \mathcal{L}(\boldsymbol{s}_i \mid (\boldsymbol{\mu}_m^{(a)}, \boldsymbol{w}_m^{(a)}, \widehat{\boldsymbol{\theta}}_{R,m})).$$

5:      Sample cluster labels $\boldsymbol{c}$ using full conditional distribution:

$$p(c_i = m \mid \cdots) \propto \begin{cases} r_m^{(a)} \mathcal{L}(\boldsymbol{s}_i \mid \boldsymbol{\mu}_m^{(a)}, \boldsymbol{w}_m^{(a)}, \widehat{\boldsymbol{\theta}}_{R,m}) & \text{for } m = 1, ..., k, \\ r_m^{(na)} \mathcal{L}(\boldsymbol{s}_i \mid \boldsymbol{\mu}_m^{(na)}, \boldsymbol{w}_m^{(na)}, \widehat{\boldsymbol{\theta}}_{R,m}) & \text{for } m = k+1, ..., k+l. \end{cases}$$

6:      Update ancillary variable $u$ by $u \sim \text{Gamma}(N, \frac{1}{t})$.
7:      Increment $j$.
8: **end while**

---

the posterior computation of $\boldsymbol{\pi}$ factorizable and gets rid of the sum-to-one constraint imposed on $\{\pi_m\}_{m=1}^{M}$, significantly simplifying the subsequent MCMC simulation process.

In summary, the joint posterior density function becomes

$$p(M, \boldsymbol{\Theta}, \boldsymbol{c}, \boldsymbol{r}, u \mid \boldsymbol{S}) \propto p(\boldsymbol{U}) \prod_{m=1}^{M} p(\boldsymbol{w}_m) p(r_m) \prod_{n=1}^{N} \pi_{c_n} \mathcal{L}(\boldsymbol{s}_n \mid \boldsymbol{\theta}_{S,c_n}, \widehat{\boldsymbol{\theta}}_{R,c_n}) \frac{p(u \mid t)}{t^N}, \quad (8)$$

where $p(\boldsymbol{U}) := p(M)p(\boldsymbol{U} \mid M)$ is the DPP prior.

Since the number of clusters changes dynamically as these algorithms proceed, it is helpful to further partition model parameters into parameters of allocated clusters and those of non-allocated clusters when applying posterior sampling. In particular, we partition $\boldsymbol{U}$ into two sets according to cluster allocations $\boldsymbol{c}$: one comprising cluster centers currently used for data allocation, denoted as $\boldsymbol{U}^{(a)} = \{\boldsymbol{\mu}_{c_1}, \ldots, \boldsymbol{\mu}_{c_n}\}$, and the other containing cluster centers not involved in the allocation, denoted as $\boldsymbol{U}^{(na)} = \boldsymbol{U} \backslash \boldsymbol{U}^{(a)}$. Note that the product measure $\mathrm{d}\boldsymbol{\mu} \times \mathrm{d}\boldsymbol{\mu}$ in $\Omega \times \Omega$ lifted by the map $(\boldsymbol{x}, \boldsymbol{y}) \mapsto \boldsymbol{x} \cup \boldsymbol{y}$ results in the measure $\mathrm{d}\boldsymbol{\mu}$, so the prior density of $(\boldsymbol{U}^{(a)}, \boldsymbol{U}^{(na)})$ is equivalent to $p(\boldsymbol{U}^{(a)}, \boldsymbol{U}^{(na)}) = p(\boldsymbol{U}^{(a)} \cup \boldsymbol{U}^{(na)})$, which follows the DPP density. $\boldsymbol{W}$ and $\boldsymbol{r}$ are partitioned in the same way. As $(\boldsymbol{U}, \boldsymbol{\pi}, \boldsymbol{W}, \boldsymbol{c})$ and $(\boldsymbol{U}^{(a)}, \boldsymbol{r}^{(a)}, \boldsymbol{W}^{(a)}, \boldsymbol{U}^{(na)}, \boldsymbol{r}^{(na)}, \boldsymbol{W}^{(na)}, \boldsymbol{c})$ are in a one-to-one correspondence,

we can refer to Eq. (8) and obtain the posterior of $(M, \boldsymbol{U}^{(a)}, \boldsymbol{r}^{(a)}, \boldsymbol{W}^{(a)}, \boldsymbol{c}, \boldsymbol{U}^{(na)}, \boldsymbol{r}^{(na)}, \boldsymbol{W}^{(na)}, u)$:

$$
\begin{aligned}
&p(M, \boldsymbol{U}^{(a)}, \boldsymbol{r}^{(a)}, \boldsymbol{W}^{(a)}, \boldsymbol{c}, \boldsymbol{U}^{(na)}, \boldsymbol{r}^{(na)}, \boldsymbol{W}^{(na)}, u | \boldsymbol{S}) \\
&\propto p(\boldsymbol{U}^{(a)} \cup \boldsymbol{U}^{(na)}) \Big[ \prod_{m=1}^{k} p(\boldsymbol{w}_m^{(a)}) p(r_m^{(a)}) (r_m^{(a)})^{n_m} \prod_{i:c_i=m} \mathcal{L}(\boldsymbol{s}_i \mid \boldsymbol{\mu}_m^{(a)}, \boldsymbol{w}_m^{(a)}, \widehat{\boldsymbol{\theta}}_{R,m}) \Big] \\
&\quad \times \Big[ \prod_{m=1}^{l} p(\boldsymbol{w}_m^{(na)}) p(r_m^{(na)}) \Big] p(u \mid t) \frac{1}{t^N},
\end{aligned}
\tag{9}
$$

where $n_m$ is the number of sequences allocated to the $m$-th component, $k$ denotes the cardinality of allocated clusters, and $l$ denotes that of non-allocated ones. $r_m^{(a)}$ and $r_m^{(na)}$ denote the allocated and non-allocated unnormalized weight, respectively. $p(u \mid t) = \frac{u^{N-1}}{(N-1)!} e^{-ut} t^N$.

Our posterior inference algorithm follows the principle of conditional Gibbs sampler [17]. We split all parameters into three groups: an allocated block $(\boldsymbol{U}^{(a)}, \boldsymbol{r}^{(a)}, \boldsymbol{W}^{(a)})$, a non-allocated block $(\boldsymbol{U}^{(na)}, \boldsymbol{r}^{(na)}, \boldsymbol{W}^{(na)})$, and remaining parameters $\{\boldsymbol{c}, u\}$, and update them in an alternating scheme. The posterior sampling procedure of TP$^2$DP$^2$ is summarized in Algorithm 1. More detailed derivations are elaborated in Appendix.

## 4   Experiments

Our model is compatible with various TPP backbones, which can detect clusters and fit event sequence data originating from a mixture of hybrid TPPs. To verify our claim, we generate a set of event sequences based on five different TPPs, including 1) Homogeneous Poisson process, 2) Inhomogeneous Poisson process, 3) Self-correcting process, 4) Hawkes process, and 5) Neural Hawkes process [15]. Based on the sequences, we construct three datasets with the number of mixture components ranging from three to five. For each dataset, we learn a mixture model of TPPs and set the backbone of the TPPs to be 1) the classic Hawkes process [11], 2) the recurrent marked temporal point process (RMTPP) [8], and 3) the Transformer Hawkes process (THP) [29], respectively. The learning methods include the variational EM of Dirichlet mixture model (Dirichlet Mixture) [27] and our TP$^2$DP$^2$. The results in Table 1 show that our method achieves competitive performance. Especially when the backbone is Hawkes process, applying our method leads to notable improvements in purity [23] and ARI [20], which means that our method is more robust to the model misspecification issue. In addition, learning RMTPP and THP by our method results in the best performance when $K_{GT} = 5$, showcasing TP$^2$DP$^2$'s adaptability to complex event sequences. More experiments are in Appendix.

Table 1: Experimental results on synthetic mixture of hybrid point processes datasets.

| Backbone | Method | $K_{GT} = 3$ | | $K_{GT} = 4$ | | $K_{GT} = 5$ | |
|---|---|---|---|---|---|---|---|
| | | Purity | ARI | Purity | ARI | Purity | ARI |
| Hawkes | Dirichlet Mixture | $0.678_{\pm 0.134}$ | $0.622_{\pm 0.097}$ | $0.620_{\pm 0.120}$ | $0.564_{\pm 0.126}$ | $0.574_{\pm 0.045}$ | $\mathbf{0.545_{\pm 0.046}}$ |
| | TP$^2$DP$^2$ | $\mathbf{0.884_{\pm 0.009}}$ | $\mathbf{0.745_{\pm 0.052}}$ | $\mathbf{0.739_{\pm 0.004}}$ | $\mathbf{0.626_{\pm 0.008}}$ | $\mathbf{0.603_{\pm 0.008}}$ | $0.538_{\pm 0.013}$ |
| RMTPP | Dirichlet Mixture | $\mathbf{0.983_{\pm 0.112}}$ | $\mathbf{0.972_{\pm 0.124}}$ | $0.751_{\pm 0.131}$ | $\mathbf{0.712_{\pm 0.213}}$ | $0.708_{\pm 0.030}$ | $0.633_{\pm 0.027}$ |
| | TP$^2$DP$^2$ | $0.974_{\pm 0.073}$ | $0.971_{\pm 0.109}$ | $\mathbf{0.753_{\pm 0.003}}$ | $0.708_{\pm 0.014}$ | $\mathbf{0.732_{\pm 0.024}}$ | $\mathbf{0.674_{\pm 0.017}}$ |
| THP | Dirichlet Mixture | $0.941_{\pm 0.093}$ | $0.870_{\pm 0.201}$ | $0.746_{\pm 0.007}$ | $\mathbf{0.666_{\pm 0.038}}$ | $0.610_{\pm 0.007}$ | $0.559_{\pm 0.043}$ |
| | TP$^2$DP$^2$ | $\mathbf{0.980_{\pm 0.035}}$ | $\mathbf{0.897_{\pm 0.110}}$ | $\mathbf{0.749_{\pm 0.002}}$ | $0.652_{\pm 0.007}$ | $\mathbf{0.650_{\pm 0.007}}$ | $\mathbf{0.600_{\pm 0.020}}$ |

## 5   Conclusion

In this paper, we propose the Bayesian mixture model TP$^2$DP$^2$ for event sequence clustering. It is shown that TP$^2$DP$^2$ could flexibly integrate various parametric TPPs including the neural network-based TPPs as components, achieve satisfying event sequence clustering results and produce more separated clusters. In the future, we plan to study the impact of alternative repulsive priors on event sequence clustering, and develop event sequence clustering methods in high-dimensional and spatio-temporal scenarios.

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

## A  Redundant Cluster Generation Problem in Traditional Mixture Models of Temporal Point Processes

In recent years, researchers have built mixture models of TPPs to tackle the event sequence clustering problem [23, 22, 27]. However, these models often suffer from overfitting during training, leading to excessive cluster generation with a lack of diversity. We illustrate this issue in the left panel of Figure 2.

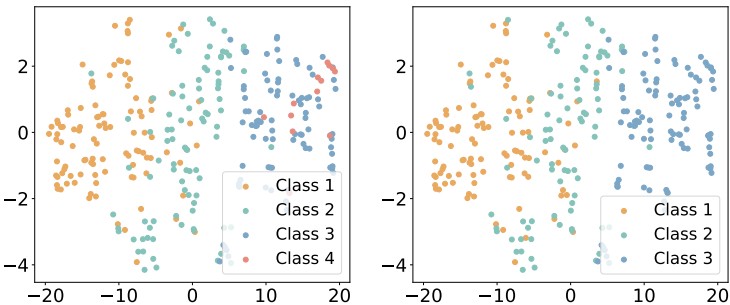

Figure 2: The t-SNE plots [19] of RMTPP's event sequence embeddings [8] for a synthetic dataset with three clusters. NTPP-MIX [27] (left) produces four clusters wrongly, while our $\text{TP}^2\text{DP}^2$ (right) leads to the clustering results matching well with the ground truth.

## B  DPP Construction

DPP prior is introduced to the central parameter $\boldsymbol{U}$ to mitigate the overfitting problem and diversify the cluster result. We leverage the spectral density approach to approximate the DPP density. For a DPP shown in Eq. (4), its kernel function has a spectral representation $\kappa(\boldsymbol{\mu}_i, \boldsymbol{\mu}_j) = \sum_{i=1}^{\infty} \lambda_i \phi_i(\boldsymbol{\mu}_i)\overline{\phi_i(\boldsymbol{\mu}_j)}$, in which each eigenfunction could be approximated via the Fourier expansion and eigenvalues are specified by the spectral distribution, as defined in [14]. In this way, the DPP density approximation is

$$p(\boldsymbol{U} \mid M) \approx \exp\left(|\mathcal{R}| - D_{\text{app}}\right) \det\{\tilde{\boldsymbol{K}}\}(\boldsymbol{\mu}_1, \cdots, \boldsymbol{\mu}_M),$$

where $\tilde{\kappa}(\boldsymbol{\mu}_i, \boldsymbol{\mu}_j) = \sum_{\boldsymbol{z} \in \mathbb{Z}^q} \tilde{\varphi}(\boldsymbol{z}) e^{2\pi i \boldsymbol{z} \cdot (\boldsymbol{\mu}_i - \boldsymbol{\mu}_j)}$, $D_{\text{app}} = \sum_{\boldsymbol{z} \in \mathbb{Z}^q} \log(1 + \tilde{\varphi}(\boldsymbol{z}))$, $\{\boldsymbol{\mu}_1, \cdots, \boldsymbol{\mu}_M\} \subset \mathcal{R}$, $|\mathcal{R}|$ is the volume of the range of the parameter space, $\tilde{\varphi}(\boldsymbol{z}) = \varphi(\boldsymbol{z})/(1 - \varphi(\boldsymbol{z}))$, $\mathbb{Z}^q$ is $q$-dimensional integer lattice, and $\varphi$ is the spectral distribution.

## C  Derivation of the Posterior Sampling Method of $\text{TP}^2\text{DP}^2$

Our posterior inference algorithm follows the principle of conditional Gibbs sampler [17]. We split all parameters into three groups: an allocated block $(\boldsymbol{U}^{(a)}, \boldsymbol{r}^{(a)}, \boldsymbol{W}^{(a)})$, a non-allocated block $(\boldsymbol{U}^{(na)}, \boldsymbol{r}^{(na)}, \boldsymbol{W}^{(na)})$, and remaining parameters $\{\boldsymbol{c}, u\}$, and update them in an alternating scheme.

**Sampling non-allocated variables:** We begin with sampling $\boldsymbol{U}^{(na)}$ from its conditional density:

$$
\begin{aligned}
&p(\boldsymbol{U}^{(na)} \mid \boldsymbol{U}^{(a)}, \boldsymbol{r}^{(a)}, \boldsymbol{W}^{(a)}, \boldsymbol{c}, u, \boldsymbol{S}) \\
&= \iint p(\boldsymbol{U}^{(na)}, \boldsymbol{r}^{(na)}, \boldsymbol{W}^{(na)} \mid \cdots) \, \mathrm{d}\boldsymbol{r}^{(na)} \, \mathrm{d}\boldsymbol{W}^{(na)} \\
&\propto \iint p(\boldsymbol{U}^{(a)} \cup \boldsymbol{U}^{(na)}) \Big[ \prod_{m=1}^{l} p(\boldsymbol{w}_m^{(na)}) p(r_m^{(na)}) \Big] \\
&\quad \times p(u|t) \frac{1}{t^N} \, \mathrm{d}\boldsymbol{r}^{(na)} \, \mathrm{d}\boldsymbol{W}^{(na)} = p(\boldsymbol{U}^{(a)} \cup \boldsymbol{U}^{(na)}) \psi(u)^l,
\end{aligned}
\tag{10}
$$

where "$\cdots$" denotes variables excluding the target variable to be sampled, together with all the sample sequences $\boldsymbol{S}$, and henceforth. $p(\boldsymbol{U}^{(a)} \cup \boldsymbol{U}^{(na)})$ is the DPP density. The second term

$\psi(u)^l = \int[\prod_{m=1}^l \exp(-ur_m^{(na)})p(r_m^{(na)})]\mathrm{d}\boldsymbol{r}^{(na)}$, due to the fact that

$$p(u \mid t) = \frac{u^{N-1}}{(N-1)!}e^{-ut}t^N = \frac{t^N u^{N-1}}{(N-1)!}\Big[\prod_{m=1}^k e^{-ur_m^{(a)}}\Big]\Big[\prod_{m=1}^l e^{-ur_m^{(na)}}\Big]. \qquad (11)$$

Applying Birth-and-death Metropolis-Hastings algorithm [10], we sample $\boldsymbol{U}^{(na)}$ and determine the final number of clusters accordingly.

We then sample $\boldsymbol{r}^{(na)}$ and $\boldsymbol{W}^{(na)}$ using the classical Metropolis-Hastings algorithm. The cardinality of non-allocated variables (i.e., $l$) is determined by the size of $\boldsymbol{U}^{(na)}$, so we have

$$p(\boldsymbol{r}^{(na)} \mid \cdots) \propto \prod_{m=1}^l p(r_m^{(na)})e^{-ur_m^{(na)}},$$
$$p(\boldsymbol{W}^{(na)} \mid \cdots) \propto \prod_{m=1}^l p(\boldsymbol{w}_m^{(na)}). \qquad (12)$$

**Sampling allocated variables:** The allocated central parameter $\boldsymbol{U}^{(a)}$ is sampled from

$$p(\boldsymbol{U}^{(a)} \mid \cdots) \propto p(\boldsymbol{U}^{(a)} \cup \boldsymbol{U}^{(na)}) \prod_{m=1}^k \prod_{i:c_i=m} \mathcal{L}(\boldsymbol{s}_i \mid (\boldsymbol{\mu}_m^{(a)}, \boldsymbol{w}_m^{(a)}, \widehat{\boldsymbol{\theta}}_{R,m}), \qquad (13)$$

where the $p(\boldsymbol{U}^{(a)} \cup \boldsymbol{U}^{(na)})$ is again governed by the DPP. Subsequently, we sample $\boldsymbol{r}^{(a)}$ from its full conditional using the Metropolis-Hastings algorithm:

$$p(\boldsymbol{r}^{(a)} \mid \cdots) \propto \prod_{m=1}^k p(r_m^{(a)})(r_m^{(a)})^{n_m} \exp(-ur_m^{(a)}). \qquad (14)$$

The $\boldsymbol{W}^{(a)}$'s full conditional is:

$$p(\boldsymbol{W}^{(a)} \mid \cdots) \propto \prod_{m=1}^k p(\boldsymbol{w}_m^{(a)}) \prod_{i:c_i=m} \mathcal{L}(\boldsymbol{s}_i \mid (\boldsymbol{\mu}_m^{(a)}, \boldsymbol{w}_m^{(a)}, \widehat{\boldsymbol{\theta}}_{R,m})). \qquad (15)$$

As $\boldsymbol{W}^{(a)}$ represents all the allocated parameters of the point process model to be inferred, excluding $\boldsymbol{\mu}$, it may still exhibit high dimensionality. To align with our framework and boost convergence, we leverage the stochastic gradient Langevin dynamics [21] when sampling $\boldsymbol{W}^{(a)}$. The proposed update for each $\boldsymbol{w}_m^{(a)}$ is provided by:

$$\Delta\boldsymbol{w}_m^{(a)} = \eta_j + \frac{\epsilon_j}{2}\Big(\nabla \log p(\boldsymbol{w}_m^{(a)}) + \frac{n_m}{n_*} \sum_{c_i=m} \nabla \log \mathcal{L}(\boldsymbol{s}_i \mid \boldsymbol{\mu}_m^{(a)}, \boldsymbol{w}_m^{(a)}, \widehat{\boldsymbol{\theta}}_{R,m})\Big), \qquad (16)$$

where $j$ is the counting number of iterations, $\eta_j \sim N(0, \epsilon_j)$, $\epsilon_j$ is the step size at the $j$-th iteration which is set to decay towards zero, and $n_*$ in the above equation is the number of selected sequences from each cluster to perform stochastic approximation. $\nabla \log \mathcal{L}(\boldsymbol{s}_i \mid \boldsymbol{\mu}_m^{(a)}, \boldsymbol{w}_m^{(a)}, \widehat{\boldsymbol{\theta}}_{R,m})$ is calculated through the automatic differentiation [2].

**Sampling $\boldsymbol{c}$ and $u$:** Each cluster label $c_i$ is sampled from

$$p(c_i = m \mid \cdots) \propto \begin{cases} r_m^{(a)}\mathcal{L}(\boldsymbol{s}_i \mid \boldsymbol{\mu}_m^{(a)}, \boldsymbol{w}_m^{(a)}, \widehat{\boldsymbol{\theta}}_{R,m}) & \text{for } m = 1, ..., k, \\ r_m^{(na)}\mathcal{L}(\boldsymbol{s}_i \mid \boldsymbol{\mu}_m^{(na)}, \boldsymbol{w}_m^{(na)}, \widehat{\boldsymbol{\theta}}_{R,m}) & \text{for } m = k+1, ..., k+l. \end{cases} \qquad (17)$$

Note that after this step, there is a positive probability that $c_i > k$ for certain indices $i$, indicating that some initially non-allocated components become allocated, and vice versa—some initially allocated components become non-allocated. Consequently, a relabeling of $(\boldsymbol{U}^{(a)}, \boldsymbol{r}^{(a)}, \boldsymbol{W}^{(a)}, \boldsymbol{U}^{(na)}, \boldsymbol{r}^{(na)}, \boldsymbol{W}^{(na)})$ and $\boldsymbol{c}$ is performed, ensuring that $\boldsymbol{c}$ takes values within the set $\{1, \ldots, k\}^N$. Thus, $k$ may either increase or decrease or remain unchanged after the relabeling step. Finally, we sample $u$ from a gamma distribution with a shape parameter of $N$ and an inverse scale parameter of $t$.

## D    Experiments

To comprehensively evaluate the effectiveness of our TP$^2$DP$^2$ model and its inference algorithm, we test our method on both synthetic and real-world datasets and compare it with state-of-the-art event sequence clustering methods. For each method, we evaluate its clustering performance by clustering purity [23] and adjusted rand index (ARI) [20] and its data fitness by the expected log-likelihood per event (ELL) [24]. In addition, we report the expected posterior value of the number of clusters ($M$) in real-world dataset, which reveals the inferred number of components given data. The code for TP$^2$DP$^2$ is publicly available at `https://anonymous.4open.science/r/TP2DP2/`.

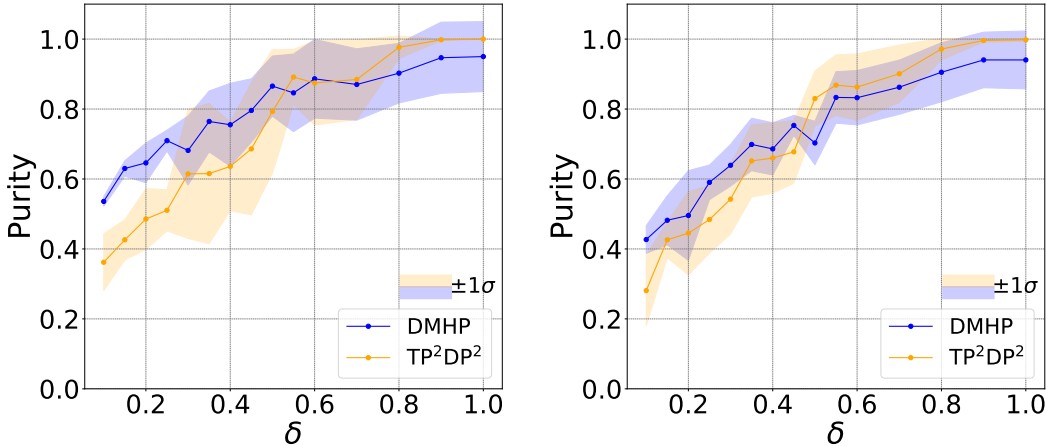

Figure 3: The means and standard deviations of clustering purity obtained by DMHP and $TP^2DP^2$ with different $\delta$. The left panel is the result when the ground truth cluster number $K_{GT} = 4$, and the right is the result of $K_{GT} = 5$.

### D.1 Experiments on Mixtures of Hawkes Processes

We first investigate the clustering capability of $TP^2DP^2$ and demonstrate the rationality of DPP priors on the synthetic datasets generated by mixture models of Hawkes processes, in which each Hawkes process generates 100 event sequences with 3 event types. All Hawkes processes apply the same triggering function, and their base intensities are set to $\boldsymbol{\mu}_m = (0.5 + \delta_m)\mathbf{1}_3$, where $\delta_m = \delta \cdot (m - 1)$ for $m \in \{1, 2, 3, \cdots, K_{GT}\}$, where $K_{GT}$ denotes the true number of clusters. In other words, these Hawkes processes exhibit distinct temporal patterns because of their different base intensities. The experiments are carried out for $K_{GT} = 4, 5$ for multiple datasets, each dataset having different $\delta$ values, $\delta \in \{0.1, 0.15, 0.2, 0.25, 0.3, \cdots, 1\}$.

We compare our $TP^2DP^2$ with the Dirichlet mixture model of Hawkes processes (**DMHP**) learned by the variational EM algorithm [23]. For a fair comparison, we use the same Hawkes process backbone as in DMHP, ensuring identical parametrization, and we consider all model parameters in the Bayesian inference phase. For each method, we initialize the number of clusters randomly in the range $[K_{GT} - 1, K_{GT} + 1]$. The averaged results in five trials are presented in Figure 3.

When the disparity in the true base intensity among different point processes is minimal, the inherent distinctions within event sequences are not apparent, as shown in Figure 4. In this case, $TP^2DP^2$ tends to categorize these event sequences into fewer groups than the ground truth, resulting in a relatively modest purity when $\delta$ is small. As $\delta$ increases, $TP^2DP^2$ exhibits increasingly robust clustering capabilities, with means consistently outperforming DMHP when $\delta > 0.55$.

In addition, we examine the posterior distribution of base intensity parameters when both algorithms converge. At $\delta = 0.6$, box plots in Figure 5 depict posterior estimations of base intensities for the first two clusters (the ground truth base intensities are 0.5 and 1.1). It is noteworthy that DMHP consistently underestimates the true values in all trials due to multiple times of approximations in its learning algorithm, and DMHP shows marginal disparity between clusters. In contrast, $TP^2DP^2$ better captures the true base intensity values, meantime exhibiting greater dispersion between clusters compared with DMHP. Similar patterns are also observed in other datasets.

### D.2 Experiments on Mixtures of Hybrid TPPs

In experiments on mixtures of hybrid TPPs, we further investigate the effect of incorporating DPP prior to different parameters in the models, and the results are shown in Table 2. In this experiment, we aim to verify adding DPP priors to central parameters of TPPs would lead to superior performance. For Hawkes Process, the base intensity reflects the average level of event occurrence rate, and is considered the most crucial for analyzing the feature of corresponding event sequences [11]. For

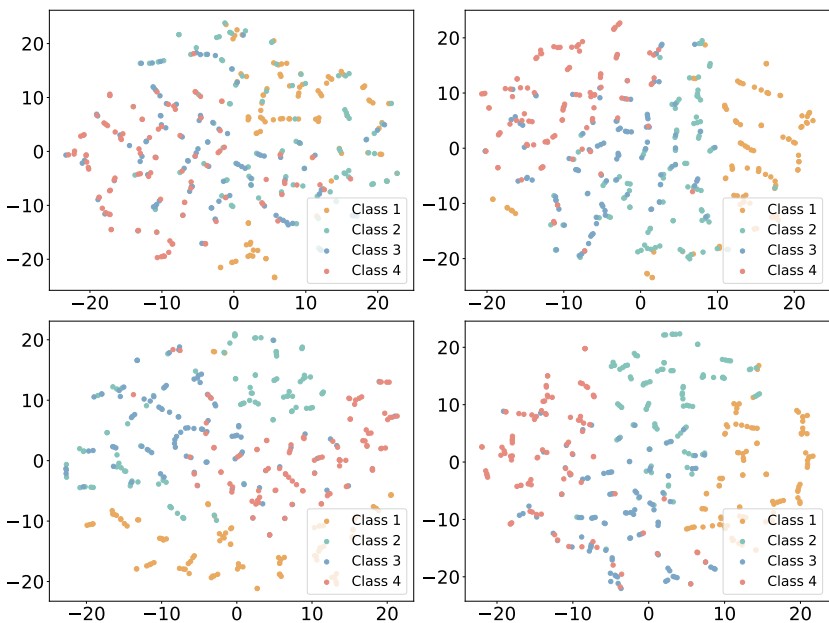

Figure 4: The t-SNE plot of the ground truth distribution for the synthetic mixture of Hawkes processes datasets with $\delta$ values of 0.2 (upper left), 0.4 (upper right), 0.6 (lower left), and 0.8 (lower right).

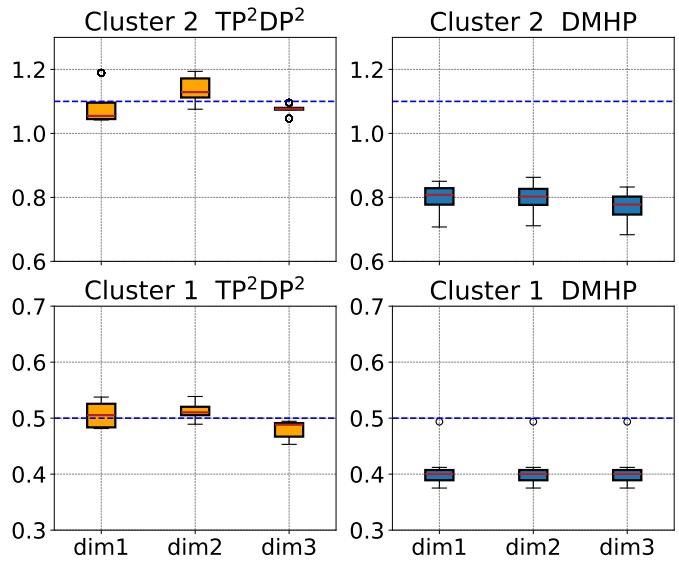

Figure 5: The base intensity $\mu$ of first two clusters learned by two methods across 5 random trials. The dotted line represents the ground truth $\mu$ in two clusters.

example, in the field of seismology, the base intensity specifically relates to background seismicity that needs special attention. Thus, for the event sequence clustering task, we also add DPP priors to the base intensity of the Hawkes process components, and find this yields the best clustering performance. For both RMTPP and THP, adding DPP prior to the output linear layer bias achieves the best purity and ARI scores, which are also consistently higher than the Dirichlet mixture frameworks or models without DPP priors. According to the architecture of these two neural point processes, the bias term of the last output layer has a direct impact on the estimated intensity function. This experimental result shows the effectiveness of applying DPP to the parameters that play a decisive role in intensity.

Table 2: Experimental results on the synthetic mixture of hybrid point processes dataset ($K_{GT} = 4$) when adding DPP prior to different parts of Hawkes processes or the bias of different layers in NTPPs. None denotes we do not impose DPP prior.

| Model | Layer | Purity | ARI |
|---|---|---|---|
| | None | 0.702 | 0.648 |
| Hawkes | Diagonal Elements of Infectivity Matrix | 0.655 | 0.572 |
| | Base Intensity | **0.739** | **0.626** |
| | None | 0.750 | 0.679 |
| RMTPP | Time Embedding Layer | 0.747 | 0.664 |
| | Output Layer | **0.753** | **0.708** |
| | None | 0.722 | 0.605 |
| | Post-attention Feedforward Layer 1 | 0.740 | 0.630 |
| | Post-attention Feedforward Layer 2 | 0.738 | 0.610 |
| THP | Post-attention Feedforward Layer 3 | 0.745 | 0.647 |
| | Post-attention Feedforward Layer 4 | 0.748 | 0.647 |
| | Output Layer | **0.749** | **0.652** |

Table 3: Experimental results on real-world datasets.

| Backbone | Method | Amazon | | BookOrder | |
|---|---|---|---|---|---|
| | | ELL | $M$ | ELL | $M$ |
| Hawkes | Dirichlet Mixture | -2.355 | 5.0 | **4.832** | 3.6 |
| | $\text{TP}^2\text{DP}^2$ | **-2.352** | 5.0 | 4.810 | 3.0 |
| RMTPP | Dirichlet Mixture | -2.251 | 3.8 | 5.613 | 2.6 |
| | $\text{TP}^2\text{DP}^2$ | **-2.052** | 3.0 | **5.624** | 2.2 |
| THP | Dirichlet Mixture | 1.629 | 3.0 | 5.814 | 2.6 |
| | $\text{TP}^2\text{DP}^2$ | **1.631** | 2.8 | **5.981** | 2.4 |

## D.3 Experiments on Real-World Datasets

To examine the performance of our method on real-world data, we use the following two benchmark datasets: 1) Amazon [25]. This dataset comprises time-stamped user product review events spanning from January 2008 to October 2018, with each event capturing the timestamp and the category of the reviewed product. Data is pre-processed according to the procedure in [24]. The final dataset consists of 5,200 most active users with 16 distinct event types, and the average sequence length is 70. 2) BookOrder [2]. This book order dataset comprises 200 sequences, with two event types in each sequence. The sequence length varies from hundreds to thousands.

To ensure fairness, hyperparameters of the Dirichlet mixture models are tuned first and we intentionally make each backbone TPP model in $\text{TP}^2\text{DP}^2$ smaller or equivalent in scale compared to those of the Dirichlet mixture framework, which means that all hyperparameters related to the backbone structure, such as hidden size, number of layers, and number of heads within the $\text{TP}^2\text{DP}^2$ framework are set to be less than or equal to their corresponding counterparts in the Dirichlet mixture framework. In this case, if $\text{TP}^2\text{DP}^2$ model achieves a higher log-likelihood with fewer cluster numbers, it indicates that our method is better at capturing the characteristics of the data and provides a better fit. Table 3 summarizes the average results for different models in five trials.

In the experiment on the real-world dataset, the Dirichlet Mixture framework performs generally worse than $\text{TP}^2\text{DP}^2$ in both dataset, but the number of posterior cluster numbers inferred by the Dirichlet Mixture framework is generally larger. This reflects that $\text{TP}^2\text{DP}^2$ moderately reduces the number of clusters to obtain more dispersed components without sacrificing much fitting capability.

---

[2]`https://ant-research.github.io/EasyTemporalPointProcess/user_guide/dataset.html`

