# OpenReview forum: "TP$^2$DP$^2$: A Bayesian Mixture Model of Temporal Point Processes with Determinantal Point Process Prior"
_NeurIPS.cc/2024/Workshop/BDU — NeurIPS BDU Workshop 2024 Poster_

### Official Review · Reviewer_uk5b · 2024-09-25

**Rating:** 6
**Confidence:** 2

**Review:**

The paper proposes to incorporate the repulsive determinantal point process (DPP) prior into a Bayesian mixture model for temporal point processes. The authors develop a specialized conditional Gibbs posterior sampling algorithm to perform Bayesian inference on selected model parameters. The experiments show the proposed method can produce moderately fewer yet more diverse mixture components and perform competitively on evaluation metrics.

The paper is well-written, with a clear motivation and a solid technical foundation. I also appreciate the authors providing an anonymous code repository for reference, which enhances transparency and reproducibility.

A detailed runtime analysis of the posterior inference algorithm would strengthen the authors' claim regarding its computational efficiency. Including a comparison with existing methods in terms of runtime would provide a more complete picture. In particular, I imagine the number of selected parameters would have a huge impact on the performance.

Minor formatting issues in the reference list, e.g., reference [3], Mcmc -> MCMC, bayesian -> Bayesian

**Overall Assessment:** Although I have limited experience in the temporal point process domain and it's likely I did not fully understand some technical details of the paper, the paper seems to present a meaningful contribution to the field, and I lean toward acceptance.

---

### Official Review · Reviewer_nSoS · 2024-09-27
**BDU Review**

**Rating:** 4
**Confidence:** 3

**Review:**

# Overview

Thanks to the authors for their submission. While my response is fairly critical, it is given in a wholly constructive spirit.

The authors propose a mixture model for temporal point processes, including a "repulsive" prior designed to increase the diversity of clusters, and derive a conditional Gibbs sampling procedure to efficiently infer the posterior. While the paper makes some interesting original strides (the DPP priors, and deriving efficient updates), and could potentially be made improved with significant revisions, in its current form it has problems in terms of quality, significance, and clarity.

The most major problem in terms of quality is a lack of significant experimental results: the proposed model is compared against only one baseline (a Dirichlet Process Mixture Model), and although the comparisons slightly favor TP$^2$DP$^2$, it is overall not convincing: TP$^2$DP$^2$ does not consistently outperform the DPMM, and in some of the cases where it does outperform it does so only by a small margin. This could be forgiven if the authors demonstrate that TP$^2$DP$^2$ is competitive in other ways, but unfortunately does not support their claim that TP$^2$DP$^2$ produces "outstanding" results. The paper would also be improved by including more than one other baseline model (modulo backbones).

The presentation and writing of the paper could also be improved. Most importantly, if the authors make specific claims about experimental performance, then some of the experimental results (or a quantitative summary) should be included in the body of the paper. As written, all experimental results are included only in the appendix. The structure and wording of the body of the paper also need significant revision in order to be clear and persuasive.

## Pros
- Good motivation of why one might want to explore an alternative prior for TPPs (priors are important and sometimes overly-neglected!)
- The author's have clearly put significant thought into how to perform inference efficiently given limitations (e.g. in applying Bayesian subnetwork inference) which can lay the groundwork for further refinements and/or help inspire new approaches.

## Cons
- As mentioned above, experimental results are quite limited.
- Paper is unclear and needs significant restructuring and rewording.

# Suggestions

## Major
- TP$^2$DP$^2$ should be compared against more than one alternative baseline model.
- If the paper makes specific claims about the quality of experimental results, these should be backed up by at least a couple significant experimental results included in the body of the paper, or by a comprehensive and quantitative summary of results.
  - In order to make room for experimental results in the body of the paper, Algorithm 1 could be summarized by more-or-less removing all of the display-math. The full mathematical form can be given in the appendix.
  - Unfortunately, I do not think the results in the appendix are strong enough w.r.t. the baseline to qualify as "outstanding", and I am not persuaded that they are particularly significant without more evidence.
- The body of the paper, in particular Section **3 Proposed TP$^2$DP$^2$ Model & Corresponding Posterior Inference Algorithm** requires significant restructuring for clarity. I suggest:
  - Paragraph break before "Unlike existing work, in Eq (6) ..."
  - Paragraph break before "Imposing the conditional independence on the central an non-central parameters ..."
  - Paragraph break before "For prior of $\pi$, instead of directly sampling ... "
  - Paragraph break before "In summary, the joint posterior density function becomes ..."
- In Section **D.1 Experiments on Mixtures of Hawkes Processes**, you say that TP$^2$DP$^2$ "consistently outperforms DMHP when $\delta > 0.55$", but your plots do _not_ show this: At best, they show that TP$^2$DP$^2$ generally outperforms DMHP when $\delta > 0.55$ (for $K_{GT} = 5$, at least): The mean of TP$^2$DP$^2$ may be slightly above DMHP, but there is significant overlap in their standard deviations. Since the mean and SD are only estimated with 5 trials, it is probably safer to say that "TP$^2$DP$^2$ is competitive with DMHP" when $\delta > 0.55$.
- In the "Broader Impacts" section of the checklist, you state that the paper does not have societal impacts. However, in the introduction, you motivate robust sequence clustering by claiming that it could facilitate healthcare decision making: what happens if your algorithm is used in such a setting but is not as robust or accurate as expected? While this may seem like nitpicking, I think these are exactly the kinds of issues that should be interrogated when considering broader societal impacts.

## Minor
- In Appendix Section *A*, the t-SNE plots aren't very informative (even as a general illustration) since the ground-truth clusters are not very evident (to my eye). Maybe you could find a better example, where the clusters are more distinct, so that NTPP-MIX is more clearly "wrong" and TP$^2$DP$^2$ more clearly "right"? Failing that, I would suggest at least plotting the ground-truth cluster assignments in a 3rd panel, so we can easily compare both to the GT.
- You should add error estimates to the ELL metric in Table 3: For example, TP$^2$DP$^2$ outperforms a Dirichlet Mixture with THP backbone by 1.631 to 1.629, but we might expect this is within the margin of error. Probably this means running more than 5 trials to get a good estimate.
- The qualitative comparison at the end of Appendix Section **D.3 Experiments on Real-World Datasets** is not very convincing: it's not a-priori clear that the bottom two sequences, although superficially similar, _should_ belong to the same cluster. Additionally, since these are just four arbitrary sequences from hundreds, what assurance do we have that they are representative of the clusters (and thereby the model performance): maybe these examples are just "lucky" for TP$^2$DP$^2$ and "unlucky" for the DPMM.
- Minor wording issues (although I fully understand these can be very hard to spot):
  - "$p( \mathbf{W} | M) = \prod_{m=1}^{M} p(\mathbf{w}_{m})$ is the prior of non-central parameters which can be Gaussian." Do you mean to say "which we choose to be Gaussian"?
  - "Introducing $u$ could make the posterior computation of $\mathbf{\pi}$ factorizable and get rid of the sum-to-one constraint imposed on ..." $\rightarrow$ "Introducing $u$ makes the posterior computation of $\mathbf{\pi}$ factorizable and gets rid of the sum-to-one constraint imposed on ..."
  - In general, I think the paper could use a thorough check of wording for clarity, and in some places for subject/verb agreement.
    - e.g. "**Determinantal Point Processes** DPP is a ..." $\rightarrow$ "**Determinantal Point Proccesses (DPPs)** are a ..."

---

### Decision · Program_Chairs · 2024-10-09

Accept (Poster)